# Bidirectional Regulation of Sodium Acetate on Macrophage Activity and Its Role in Lipid Metabolism of Hepatocytes

**DOI:** 10.3390/ijms24065536

**Published:** 2023-03-14

**Authors:** Weiwei Li, Mingjuan Deng, Jiahui Gong, Yichao Hou, Liang Zhao

**Affiliations:** 1College of Food Science and Nutritional Engineering, China Agricultural University, Beijing 100083, China; 2Key Laboratory of Functional Dairy, Department of Nutrition and Health, China Agricultural University, Beijing 100193, China; 3Research Center for Probiotics, China Agricultural University, Sanhe 065200, China

**Keywords:** sodium acetate, macrophages, dose-dependent effects, AMPK, lipid accumulation, hepatocyte

## Abstract

Short-chain fatty acids (SCFAs) are important metabolites of the intestinal flora that are closely related to the development of non-alcoholic fatty liver disease (NAFLD). Moreover, studies have shown that macrophages have an important role in the progression of NAFLD and that a dose effect of sodium acetate (NaA) on the regulation of macrophage activity alleviates NAFLD; however, the exact mechanism of action remains unclear. This study aimed to assess the effect and mechanism of NaA on regulating the activity of macrophages. RAW264.7 and Kupffer cells cell lines were treated with LPS and different concentrations of NaA (0.01, 0.05, 0.1, 0.5, 1, 1.5, 2, and 5 mM). Low doses of NaA (0.1 mM, NaA-L) significantly increased the expression of inflammatory factors tumor necrosis factor-α (*TNF-α*), interleukin-6 (*IL-6*), and interleukin 1 beta (*IL-1β*); it also increased the phosphorylation of inflammatory proteins nuclear factor-κB p65 (NF-κB p65) and c-Jun (*p < 0.05*), and the M1 polarization ratio of RAW264.7 or Kupffer cells. Contrary, a high concentration of NaA (2 mM, NaA-H) reduced the inflammatory responses of macrophages. Mechanistically, high doses of NaA increased intracellular acetate concentration in macrophages, while a low dose had the opposite effect, consisting of the trend of changes in regulated macrophage activity. Besides, *GPR43* and/or *HDACs* were not involved in the regulation of macrophage activity by NaA. NaA significantly increased total intracellular cholesterol (TC), triglycerides (TG), and lipid synthesis gene expression levels in macrophages and hepatocytes at either high or low concentrations. Furthermore, NaA regulated the intracellular AMP/ATP ratio and AMPK activity, achieving a bidirectional regulation of macrophage activity, in which the PPARγ/UCP2/AMPK/iNOS/IκBα/NF-κB signaling pathway has an important role. In addition, NaA can regulate lipid accumulation in hepatocytes by NaA-driven macrophage factors through the above-mentioned mechanism. The results revealed that the mode of NaA bi-directionally regulating the macrophages further affects hepatocyte lipid accumulation.

## 1. Introduction

Non-alcoholic fatty liver disease (NAFLD) includes a wide range of disorders associated with fat deposition in the liver, spanning from isolated steatosis to non-alcoholic steatohepatitis (NASH) characterized by steatosis with hepatocellular injury and inflammatory changes with or without fibrosis [1]. NAFLD affects around 32.4% of the general population worldwide. NASH is the active form of NAFLD characterized by histological lobular inflammation and hepatocyte ballooning and associated with faster fibrosis progression, which affects around 1.5–6.5% of the general population [2,3]. Alarmingly, the incidence of NAFLD in adults and children has been continuously increasing due to ongoing epidemics of obesity [4,5]. Moreover, NASH is among the most common causes of liver cirrhosis and hepatocellular carcinoma, ultimately leading to liver failure [6,7,8]. Thus far, there is still no effective treatment for NAFLD [2,9].

Over the past decade, the essential role of innate immunity in developing hepatic steatosis and NASH has been extensively validated [10,11,12]. As one of the most studied cell types in innate immunity, macrophages have drawn particular attention because macrophage pro-inflammatory activation is strongly associated with hepatic steatosis and inflammation. When pro-inflammatory activation increases, macrophages can generate mediators that trigger or exacerbate hepatocyte inflammatory responses and fat metabolic dysregulation [10,11,12]. Resident tissue macrophages (Kupffer cells) along with monocytes (bone-marrow-derived macrophage, BMDM) recruited from the bone marrow are also important drivers of inflammatory and tissue regenerative responses [13]. Following tissue injury, large numbers of inflammatory monocytes (macrophage precursors) are recruited from the bone marrow via chemokine gradients and various adhesion molecules, and these recruited cells often greatly exceed the population of tissue-resident macrophages [14]. To date, several regulators, such as c-Jun N-terminal kinase 1 (JNK1), Period 1 and 2, and adenosine 2A receptor, have been found to alter the inflammatory status of macrophages, which in turn interact with hepatocytes to protect against or contribute to hepatic steatosis and inflammation [15,16,17]. These findings demonstrate the importance of the innate immune system, particularly macrophages, in the pathophysiology of NAFLD.

Accumulating evidence shows a close relationship between the gut and NAFLD [18,19,20]. As the liver receives most of its blood flow (70%) from intestinal vascularization, it is constantly exposed to nutrients, toxins, and gut microbiota products [18]. Moreover, the gastrointestinal tract receives a liver product in the form of bile acid [19], and this functional bidirectional correlation between the liver and gastrointestinal tract is known as the gut-liver axis (GLA) [18]. Gut microbiota is composed of trillions of microorganisms that create a symbiotic relationship with the host or reside as commensals and can execute various functions influencing human physiology and pathology. SCFAs, the main product of dietary fiber metabolized by intestinal flora, are considered important substances related to the gut-liver axis [20,21]. SCFAs might have an essential role in regulating inflammation and metabolism, contextually exerting regulative effects on NAFLD. Acetate, propionate, and butyrate are the main SCFAs in the human gut, accounting for more than 90% of the total SCFAs [22]. Acetate is a two-carbon SCFA known to be used for de novo synthesis for lipids and as a signaling molecule involved in several physiologic processes, including reductions in insulin resistance and appetite, thus contributing to glucose homeostasis, control of body weight, and progress of NAFLD [21,23,24]. Meanwhile, acetate regulates immune response, which might affect NAFLD from another aspect. Acetate supplementation leads to an effective decline in methionine-choline deficient (MCD)-induced macrophage aggregations and pro-inflammatory responses [25]. Besides, acetate supplementation can effectively reduce inflammation and improve insulin sensitivity in HFD-induced diabetic mice by reducing the levels of *IL-1β* and *IL-6* [26]. In contrast, a recent study emphasized that high fecal SCFAs content can impact NAFLD progression by maintaining intestinal low-grade inflammation [22]. Moreover, the principal hepatic metabolite of ethanol, acetate incubation can simulate the effect of ethanol, and increase the production of cellular inflammatory factors by increasing the synthesis of Acyl-CoA Synthetase Short Chain Family Member 2 (ACSS2), the consequent enhancement of the inflammatory response in macrophages [27]. While acetate has been posited to closely related inflammatory response, its role in inflammatory liver pathogenesis remains unclear.

The total SCFAs concentration in the colon’s lumen decreases progressively from the proximal to the distal end from 70~140 mmol/L to 20~70 mmol/L, respectively [28], with the ratio of acetate, propionate, and butyrate in the colon of 60:25:15 [29]. SCFAs are absorbed by the colonic epithelial cells, where they pass the portal vein; however, they are mostly metabolized by the hepatocytes in the liver. The concentration range of acetate in humans has a large variation. There is evidence that the acetate is closely related to the activation state of macrophages [30], and our previous studies suggested that there may be a dose effect for acetate to work and that high doses of sodium acetate might be required [25,31]. Currently, there is a lack of research and exploration of relevant mechanisms for the inconsistent results of acetate on NAFLD, especially the dose-effect relationship.

The present study verified the dose effect of acetate in regulating macrophage activity of bone marrow origin and liver origin, investigated the mechanism of the bidirectional regulation of acetate in regulating macrophage activity from the perspective of signaling pathways and fatty acid synthesis, and assessed the effect of the dosage effect on fat accumulation in hepatocytes at the cellular level. Our findings revealed the role of acetate in regulating NAFLD at the cellular level.

## 2. Results

### 2.1. NaA Regulates the Inflammatory Responses of Macrophages in a Dose-Dependent Manner

A CCK-8 assay was performed to evaluate the potential cytotoxic effect of NaA on macrophages. The results showed that after treatment with the maximum concentration of NaA (10 mM) for 24 h, >80% of the macrophages (RAW264.7 and Kupffer cells) survived (Figure 1A,B), indicating that NaA did not affect cell viability.

Regarding the inflammation-modulating effects of the different doses of NaA on LPS-induced RAW264.7 and Kupffer cells activation, the gene expression level of *TNF-α*, the marker of inflammation, was analyzed by qRT-PCR. LPS treatment significantly elevated *TNF-α* in the MOD group compared to the CON group, which indicated macrophage activation by LPS. Furthermore, low doses of NaA treatment significantly promoted the overexpression of *TNF-α* in LPS-activated RAW264.7 and Kupffer cells, while high-dose NaA significantly inhibited the overexpression of *TNF-α* in LPS-activated RAW264.7 and Kupffer cells compared to the MOD group. Specifically, 0.01~0.05 mM NaA could slightly increase the expression of *TNF-α,* but there were no significant changes between these two groups and the MOD group (Figure 1C,D). Moreover, a low dose of NaA (0.1 mM) increased the levels of *TNF-α*, which were 48.7% (RAW264.7) and 66.4% (Kupffer cells) higher than those in the MOD group, respectively. In contrast, high doses of NaA (2~5 mM) reduced approximately 48% or 53% *TNF-α* levels in RAW264.7 or Kupffer cells compared to the MOD group, respectively. In addition, 0.5~1.5 mM NaA did not affect the expression of *TNF-α,* compared with the MOD group (Figure 1C,D). The results explicitly showed that NaA regulated the pro-inflammatory response of macrophages in a dose-dependent manner, while 0.1 mM NaA (NaA-L) and 2 mM NaA (NaA-H) were selected for further experiments.

### 2.2. High Dose or Low Dose of NaA Show Opposite Effects on Inflammatory Proteins and Gene Expression in Macrophages

The levels of inflammatory cytokines *IL-6* and *IL-1β* in macrophages RAW264.7 (Figure 2A,B) and Kupffer cells (Figure 2C,D) were measured by qRT-PCR. The data showed that the high levels of *IL-6* and *IL-1β* in macrophages (RAW264.7 and Kupffer cells) evoked by LPS were markedly scavenged by the post-treatment with NaA-H. Moreover, NaA-H significantly blocked the increased phosphorylation level of c-Jun and NF-κB induced by LPS in macrophages RAW264.7 (Figure 2E) and Kupffer cells (Figure 2F). In contrast, NaA-L notably exacerbated high levels of *IL-6* (Figure 2A,B) and *IL-1β* (Figure 2C,D) compared with the MOD group. Correspondingly, NaA-L also promoted phosphorylation levels of c-Jun and NF-κB, compared to the MOD group (Figure 2E,F). These data clearly demonstrated that high or low doses of NaA have opposed effects on inflammation in macrophages.

### 2.3. NaA Regulates M1 but Not M2 Macrophages

To further investigate the effects of various concentrations of NaA on the activity of macrophages, murine macrophage RAW264.7 cells were co-cultured with different doses of NaA for 24 h and then stimulated with LPS for 6 h, respectively. It is known that M1/M2 macrophage balance polarization governs the fate of inflammation [32]. So, in this study, a flow cytometric assay was used to analyze the effect on macrophage inflammatory response. The results of flow cytometry showed that the ratios of M1 macrophage polarization significantly increased after LPS induction (MOD), while NaA-H inhibited M1 macrophage polarization (from 54.6% to 48.1%) compared with the MOD group. NaA-L treatment aggravated M1 macrophages (from 54.6% to 58.1%) compared to the MOD group (Figure 3).

### 2.4. NaA Alters Intracellular Free Acetate Content by Entering Macrophages and Regulating Lipid Synthesis

Next, we examined whether *GPR43*, a promising acetate receptor [33], participates in NaA regulating RAW264.7 cell activity. The *GPR43* siRNA was used to silence its target mRNA, specifically in RAW264.7; three *siGPR43* sequences, i.e., *siGPR43-1, siGPR43-2*, and *siGPR43-3,* were designed. The qRT-PCR results showed that RAW264.7 cells had very low background expression of *GPR43* in the context of this experiment (Appendix A), and the gene silence did not affect the role of NaA. This indicated that NaA-induced activation of RAW264.7 was independent of *GPR43*. In addition to G proteins, histone deacetylase enzymes may also act as target sites of SCFAs. Since the *GPR43* receptor does not influence the function of NaA, the impact of NaA on *HDACs*, which is speculated to be involved in the beneficial effects of SCFAs, was further investigated [34]. Compared to the CON group, LPS or NaA treatment did not significantly alter the expression of *HDACs* (Appendix A). These results indicated that *GPR43* or *HDACs* might not participate in the modulation of NaA on macrophages.

On the other hand, NaA may enter into the cells and exert its role. To verify this hypothesis, we used GC-MS to detect the intracellular acetate content after RAW264.7 cells incubated with NaA. Compared to the CON and MOD groups, treatment with NaA-H caused a significant increase in the content of acetate. Intriguingly, the levels of intracellular acetate descended in the NaA-L group compared with the MOD or CON groups. The content was about 21.3 g/g prot, accounting for only 1/4 of the CON group and 1/2 of the MOD group. On the contrary, NaA-H treatment significantly increased intracellular acetate compared to the MOD or CON group (Figure 4A). We observed that the intracellular acetate content of RAW264.7 cells treated with NaA was consistent with the trend that acetate regulated RAW264.7 cell activity. These results showed that NaA might modulate RAW264.7 activation by altering intracellular acetate content.

Acetate might be used as a substrate for cell lipid synthesis, and lipid accumulation can aggravate the inflammatory response of macrophages. In order to clarify whether NaA regulated the inflammatory response of cells by affecting lipid synthesis, we determined the enrichment of RAW264.7 cells for TG, TC, lipid synthesis protein ACSS2, and lipid synthesis-related genes (*FAS*, *Scd1*, and *ACC1*). The results showed that NaA-L or NaA-H markedly increased the levels of TC or TG compared with CON and MOD groups (Figure 4B,C). Next, we investigated the role of ACSS2 in the upregulation of fat deposition in RAW264.7, where WB results showed that ACSS2 was highly expressed in the group of NaA (Figure 4D). Similarly, the mRNA levels of *Scd1*, *FAS*, and *ACC1* in RAW264.7 with LPS were significantly elevated by NaA (Figure 4E). Our findings indicated that NaA has a critical function in fat accumulation linking activation of macrophages, while both high and low doses of NaA can increase cell lipid synthesis. The inhibition of macrophage inflammatory response by NaA-H may depend on other pathways.

### 2.5. NaA Regulates Macrophage Inflammation Response Dependent on the AMPK Signaling Pathway

We wanted to investigate whether the metabolic effects of NaA might be mediated through the activation of AMPK, which is considered the target of SCFAs involved in the alleviation of metabolic disease. Our previous study showed that NaA could have a regulatory role in organismal diseases by regulating the phosphorylation of AMPK [31]. Furthermore, we used HPLC to measure the intracellular ratio of AMP/ATP. We observed that the AMP/ATP ratio was significantly decreased in the MOD group compared with the control treatment and further reduced in the NaA-L group, while it was restored in the NaA-H group. This result is consistent with the trend of inflammatory activation of macrophages. The ratio of AMP/ATP could contribute to the changes in the AMPK phosphorylation level (Figure 5A,B). To make sure whether AMPK was involved in NaA-mediated RAW264.7 inflammation, Compound C was used to block the expression of AMPK. Pre-treatment with Compound C significantly attenuated the expression of phospho-AMPK and eliminated the NaA-suppressed overexpression of phospho-c-Jun and phospho-NF-κB (Figure 5C,D). When taken together, these data suggest that NaA regulates the activation of macrophages in an AMPK-dependent manner.

### 2.6. NaA Modulates Macrophages Inflammation Activation by Activating the PPARγ-UCP2-AMPK-NF-κB-IκBα-iNOS Pathway

NaA treatment changed macrophages’ ATP concentrations and altered AMP/ATP ratios. The latter is a direct activator of AMPK [35]. Reduced ATP concentrations can result from increased mitochondrial proton leakage, leading to mitochondrial uncoupling and subsequently reduced ATP synthesis [36]. In line with this, treatment with NaA led to altered expression of UCP2 (Figure 6A), a mitochondrial uncoupling protein [36]. Next, we studied how NaA activated the UCP2-AMPK pathway. Possible candidates were PPARγ, well-known regulators of UCP2 expression, fatty acid oxidation, and whole-body lipid metabolism [35]. NaA did change the expression of PPARγ (Figure 6B), suggesting that PPARγ may be the mediating factor between NaA and the UCP2-AMPK pathway.

Since AMPK deletion leads to NF-κB activation and there is a possibility that AMPK inhibits IKK-dependent IκBα phosphorylation either directly or indirectly, we investigated if the macrophages’ activation by acetate regulation is mediated through AMPK and the specific mechanisms by which AMPK functions [37]. Subsequently, we examined the role of NaA in the activation of the transcription factor, NF-κB mediated by IκBα, which is required for the transcriptional expression of genes related to inflammation. LPS and NaA-L treatment markedly induced the phosphorylation of IκBα compared with the CON group. Nonetheless, these increases could be attenuated by treatment with 2 mM of NaA (Figure 6C), thus demonstrating that NaA regulates the activation of NF-κB regulated by IκBα.

The anti-inflammatory activity of AMPK is exerted through multiple signaling pathways, including phosphorylation and activation of inducible nitric oxide synthase (iNOS) and nitric oxide (NO) production. In addition, NO may act as an endogenous activator of AMPK, suggesting a reciprocal relationship between AMPK and iNOS [38]. Accordingly, the iNOS/NO system is likely to participate in such an acetate mechanism, which we also evaluated. The qRT-PCR results showed basal *iNOS* levels in the CON group. Treatment with NaA and LPS did change *iNOS* levels compared to the basal level of the CON group. qRT-PCR data showed that treatment with LPS and NaA-L significantly reduced the expression of *iNOS* compared with the basal level. In contrast, RAW264.7 from the NaA-H group exhibited a significant addition in *iNOS* levels compared with the NaA-L group (Figure 6D).

### 2.7. NaA-Driven Macrophage Factors Regulated Hepatocyte Fat Deposition in Co-Culture Experiment

We clearly showed that NaA could bi-directionally regulate macrophage activity. Next, we verified whether this process might have a role in NAFLD and whether it also has a two-way regulatory effect on fat accumulation in hepatocytes. First, we verified the direct effect of NaA on fat accumulation in hepatocytes. The AML-12 hepatocytes were treated with NaA in the presence of PA. The viability of AML-12 cells was not significantly reduced by incubating with either 0~10 mM NaA. However, incubation of AML-12 hepatocytes with PA medium resulted in a marked increase in hepatocyte fat deposition, and this increase was significantly enhanced upon treatment with NaA at various concentrations (0.1 and/or 2 mM). The enhancement effect of NaA was in a concentration-dependent manner from 0.1 to 2 mM (Figure 7A). In line with this, the incubation of AML-12 hepatocytes with PA and NaA led to the increased production of TC and TG, and the increase in TC and TG was also positively correlated with the concentration of NaA (Figure 7B,C). Next, we sought to examine changes in the expression of genes related to lipid metabolism. Compared with the control medium, the PA-added medium markedly increased the mRNA levels of *FATP2, FAS, ACC1, SREBP-1c,* and *Scd1*. Compared with the MOD group, NaA treatment increased the production of *FATP2, FAS, ACC1, SREBP-1c,* and *Scd1* by 53.94, 32.48, 90.06, 55.60, and 74.29% at 0.1 mM, respectively and by 128.74, 121.89, 188.63, 210.03 and 189.75% at 2 mM, respectively. Similarly, overexpression of genes was promoted, and there was a dose-response relationship from 0.1 to 2 mM (Figure 7D).

The functions of the liver can be achieved through cellular communication between hepatocytes, macrophages, and vascular endothelial cells. Because macrophages are believed to patrol systemic conditions [39], Kupffer cells, liver-resident macrophages, are likely to sense and react to inflammation. Therefore, we hypothesized that macrophages secrete molecules in response to extracellular inflammation that affects hepatocytes’ lipid accumulation. Thus, AML-12 cells were treated with a conditioned medium from RAW264.7 cells. Incubation of AML-12 hepatocytes with PA resulted in a marked increase in hepatocyte fat deposition and TC and TG content, and this increase was significantly reduced upon treatment with NaA at 2 mM concentrations. Interestingly, NaA-L exacerbated PA-induced lipid accumulation in AML-12 hepatocytes (Figure 7A–C). Concerning the expression of genes for lipogenesis, treatment with NaA-L significantly upregulated the mRNA levels of *FATP2, FAS, ACC1, SREBP-1c,* and *Scd1* compared with MOD treatment in a PA-contained medium. In contrast, incubation of AML-12 with NaA-H significantly decreased macrophage-induced overexpression of *FATP2, FAS, ACC1, SREBP-1c,* and *Scd1* (Figure 7E).

Next, we performed macrophage and hepatocyte co-cultures to further examine whether NaA facilitates macrophages’ generation of factors to promote NAFLD aspects. PA-induced hepatocyte fat deposition in NaA-L treated co-cultures of RAW 264.7 and AML-12 hepatocytes were much greater compared to the MOD group. However, this increase in hepatocyte fat deposition was not observed in NaA-H-treated co-cultures of RAW 264.7 and AML-12 hepatocytes. Conversely, NaA-H significantly reduced AML-12 hepatocytes lipid deposition through co-culture with RAW 264.7 (Figure 7A–C,F). Together with the findings presented in Figure 7E, these results strongly indicated that NaA-H could alleviate fat deposition in hepatocytes, especially lipogenesis, by inhibiting the inflammatory response of macrophages, while NaA-L exacerbated LPS-induced macrophages (RAW 264.7) inflammatory activation, which in turn aggravated AML-12 hepatocytes lipid deposition.

## 3. Discussion

The present study demonstrated that NaA could bi-directionally regulate macrophage activity. We also found that NaA entry into cells alters intracellular acetate concentration independent of *GPR44* or *HDACs*. Besides, NaA regulates the inflammatory pathways induced by LPS in macrophages by modulating AMPK-dependent IκBα/NF-κB activation. Moreover, NaA was found to regulate hepatocyte lipid accumulation through macrophage-hepatocyte interactions bi-directionally. To the best of our knowledge, this study first reported a dose-dependent bidirectional regulation of macrophage inflammatory response and mechanism by NaA.

Our results showed that NaA could regulate macrophages in both directions over a range of concentrations (0.1~2.0 mM). Importantly, these concentrations reflect the range of SCFA concentrations observed in human blood [8,30]. In addition, this study observed that NaA could increase lipid accumulation in hepatocytes at high and low doses when it acts directly on hepatocytes. Differently, we found that high and low doses of NaA inhibited or aggravated the inflammatory response of macrophages and then inhibited or aggravated the fat accumulation of hepatocytes through the interaction between macrophages and hepatocytes, showing a potential bi-directionally regulatory effect on NAFLD.

Lipid accumulation can cause cells to be more sensitive and susceptible to inflammatory responses [15,40]. Previous studies reported inconsistent effects of NaA on the inflammatory response and lipid accumulation. For example, NaA can be converted to an excess of acetyl-CoA, which increases pro-inflammatory cytokine gene histone acetylation by increased substrate concentration and *HDAC* inhibition, leading to enhanced gene expression and perpetuation of the inflammatory response of macrophages [27]. In addition, ACSS2 facilitates the consumption of extracellular acetate as an alternative carbon source and leads to lipid deposition, causing an inflammatory response in tumor cells [41,42]. On the contrary, NaA can improve cellular lipid deposition and inflammatory response by activating AMPK [31,35] and exerting an anti-inflammatory role as a *GPR43* ligand [20]. The results of this study may explain the inconsistent results from previous studies reporting that NaA improved or aggravated fatty liver with dose-dependent effects.

NaA regulates macrophage activity independently of the classical *GPR43* receptor or as an inhibitor of *HDACs* but exerts a bidirectional regulation by entering the cell and changing the intracellular acetate concentration. The biological responses of NaA on host cells result from the inhibition of *HDACs* or the activation of GPRs, such as GPR41 and *GPR43* [43]. The results in this study are consistent with the previous report, wherein SCFAs attenuated intestinal inflammation by entering Caco-2 cells [44] through monocarboxylate transporter 1 (MCT1) or sodium monocarboxylate transporter 1 (SMCT1) transport protein and acted independently of *GPR43* or *HDACs* [31]. Similarly, another study indicated that acetate was absorbed mainly by passive diffusion, accumulated inside m-ICcl2 cells, and stimulated lipid consumption in enterocytes [45]. The altering intracellular concentration of NaA demonstrated that NaA partially influenced RAW264.7 cell activation by entering the cells.

The bidirectional regulatory effect of NaA on macrophages mainly occurs by altering the intracellular free acetate content. NaA increases fatty acid synthesis by entering cells, leading to lipid deposition in macrophages and lipid accumulation in macrophages, eventually activating cellular inflammation. On the one hand, NaA can promote cellular inflammatory response by synthesizing lipids and promoting lipid accumulation in macrophages, either high dose or low dose. NaA may increase lipid generation through increased ACSS2 expression, which could be through increased substrate supply (although this acetate must be in the form of acetyl-CoA) [36]. In line with this, previous studies also demonstrated that SCFAs, especially acetate, contributed to obesity and liver steatosis as they provide approximately 10% of daily caloric consumption and might enhance nutrient absorption by promoting the expression of glucagon-like peptide 2, obesity, and liver steatosis that could trigger liver inflammation and NAFLD [35]. On the other hand, using NaA to synthesize substrates leads to changes in the actual intracellular acetate content. Low doses of NaA are mainly used to promote lipid synthesis, leading to the consumption of acetate and lower intracellular free acetate content. Accordingly, low doses of NaA are mainly used to synthesize lipids [46], and lipid accumulation contributes to the progression of inflammation and disease [47]. High doses of NaA similarly act as a substrate for lipid synthesis but can increase the intracellular free acetate content after offsetting consumption. However, a high dose of sodium acetate reduced the inflammatory response of macrophages. A previous study also showed that NaA could alleviate NASH in mice by alleviating macrophage inflammation [25] or inhibiting the inflammatory response induced by lipid deposition in hepatocytes [31,35]. Yet, whether NaA directly changes intracellular acetate concentration or regulates intracellular metabolism to change acetate concentration needs to be further verified using isotope labeling or other ways. Our results also indicated other mechanisms through which NaA inhibits macrophage inflammation, achieving an even stronger inhibitory effect than the pro-inflammatory effect of lipid synthesis. Further research is required to establish the relationship between lipids and inflammation in more detail.

AMPK is a key site in acetate’s bidirectional regulation of macrophage activity. It is identified as a crucial regulator of metabolism. Previous studies reported that enterocytes exposed to acetate induced a marked increase in phosphorylated AMPK and ameliorated lipid metabolism [48]. We previously reported that high-dose NaA treatment could affect the activation state of macrophages and hepatic astrocytes by activating AMPK. Of particular importance, it was shown that low doses of NaA had no significant effect on AMPK [25,31]. Therefore, the present study focused on investigating whether acetate’s bidirectional regulatory effects on the regulation of macrophage activity were related to AMPK. Recent reports have detected that AMPK activation is key in inflammation inhibition, particularly via the NF-kB signal pathway [49], and regulates M1/M2 macrophage polarization [36,49]. We observed that high and low NaA concentrations regulate the AMP/ATP ratio bidirectionally, controlling the phosphorylation level of AMPK. Moreover, after the inhibition of AMPK activity, the macrophage-modulating effect of NaA disappeared, confirming that AMPK was an important key site for bidirectional regulation. We observed that NaA could alter the intracellular free acetate content through entry to alter the expression of lipid synthesis genes (*Scd1, FAS,* and *ACC1*) and proteins (ACSS2), thus changing the intracellular acetate content. Furthermore, NaA could affect UCP2 expression by regulating PPARγ activity. Recent reports suggest that PPARγ is a central regulator of the role of SCFAs in regulating organismal immunity, which is also a known regulator of UCP2 expression, fatty acid oxidation, and whole-body lipid metabolism [31,35]. In addition, changes in UCP2 expression can lead to cell uncoupling, thus affecting intracellular ATP concentration [35] and altering intracellular AMP/ATP concentration, which is the main cause of AMPK activation. Hence, we found bidirectional regulation of intracellular acetate concentration and direct regulation of PPARγ by NaA. Our findings demonstrated that the PPARγ-UCP2-AMPK pathway mediated the pharmacological effect of NaA on macrophage activation.

NaA achieves a bidirectional modulatory effect on inflammation through bidirectional regulation of AMPK activity. Given that NF-κB and c-Jun are major effector molecules of NaA in LPS-treated RAW264.7 cells, we tried to investigate how these effectors stimulate the other molecules relevant to inflammatory processes. NF-κB is a multifaceted transcriptional regulator that moves into the nucleus during the activation of IκBα to regulate gene expressions related to the apoptotic and inflammatory responses [17]. AMPK has an extensive role in numerous pathways and is especially closely related to metabolic diseases [50]. In addition, it has been identified as a central modulator in macrophage function. AMPK is exerted through multiple signaling pathways, including the phosphorylation and activation of iNOS and NO production. NO may act as an endogenous activator of AMPK, suggesting a reciprocal relationship between AMPK and iNOS [38]. Moreover, NO can directly inhibit the activity of NF-κB through the phosphorylation of IKβα, an NF-κB inhibitory protein [17,38]. Our results revealed that NaA regulates the level of iNOS and the phosphorylation of IKβα mediated by AMPK to regulate the activation of NF-κB as evoked by LPS. Considering the effect of c-Jun on the NF-κB activity, many previous studies demonstrated that activated c-Jun could stimulate the transcriptional activation of NF-κB in promoting inflammation [17,38,51]. Consistent with this, our results suggest that NaA can influence the activation of c-Jun. Hence, it is conceivable that NaA harbors an important function in blocking the NF-κB-mediated apoptotic pathway by inhibiting oxidative c-Jun activation. Overall, our results suggest that AMPK is responsible for the modulation of the c-Jun and NF-κB pathways induced by NaA.

## 4. Materials and Methods

### 4.1. Materials and Reagents

RAW264.7 cells, a cell line of mouse macrophages, were purchased from the Stem Cell Bank, Chinese Academy of Sciences (Shanghai, China). Kupffer cells, a cell line of mouse liver macrophages, were purchased from BeNa Culture Collection (Beijing, China). AML-12 cells, a line of mouse hepatocytes, were purchased from Stem Cell Bank, Chinese Academy of Sciences, Dulbecco’s modified Eagle’s medium (DMEM), Roswell Park Memorial Institute (RPMI) 1640, and fetal bovine serum (FBS) were purchased from Gibco (Grand Island, NY, USA). 100 U/mL penicillin and streptomycin, Lipopolysaccharides (LPS), Enhanced Cell Counting Kit 8 Assay, and Goat anti-rabbit horseradish peroxidase-conjugated secondary antibody were from Beyotime (Beijing, China). Sodium acetate was acquired from Sigma-Aldrich (St. Louis, MO, USA). SYBR Premix Ex Taq was purchased from TaKaRa (Beijing, China). TRIzol Reagents were purchased from Ambion (Austin, TX, USA). The commercial kits for TC/TG were obtained from Nanjing Jiancheng Bioengineering Co., Ltd. (Nanjing, China). The rabbit polyclonal antibodies specific to NF-κB p65, Pp65, phosphorylated c-Jun (Ser63) (p-c-Jun), c-Jun, β-Actin, phosphorylated Adenosine5’-monophosphate activated protein kinase (pAMPK), AMPK, ACSS2, uncoupling protein 2 (UCP2), peroxisome proliferator-activated receptor γ (PPARγ), a phosphorylated inhibitor of nuclear factor kappa B (pIκBα) and IκBα were obtained from Abcam (Shanghai, China). 6/12/96-well plates were purchased from Corning (Corning, NY, USA).

### 4.2. Cell Culture and Treatment

RAW264.7 cells were cultured in DMEM containing 10% FBS and 1% 100 U/mL penicillin and streptomycin in a 5% CO_2_ incubator at 37 °C. In addition, Kupffer cells were cultured in RPMI 1640 containing 10% FBS and 1% 100 U/mL penicillin and streptomycin in a 5% CO_2_ incubator at 37 °C.

In order to study the effects of NaA on LPS-induced inflammation, RAW264.7 or Kupffer cells were seeded in 6-well plates at a density of 1 × 10^6^ cells/2 mL DMEM or RPMI 1640 medium. After achieving 70~80% confluency, the cells were subjected to 12 h of serum starvation, after which the CON group (control group) was incubated with fresh DMEM or RPMI 1640 medium, and the cells in the other group were incubated with DMEM or RPMI 1640 medium with varying concentrations of NaA (0, 0.01, 0.05, 0.1, 0.5, 1, 1.5, 2 or 5 mM) for an additional 24 h in the presence of LPS (100 ng/mL) for the last 30 min to examine inflammatory signaling or LPS (20 ng/mL) for 6 h to quantify the expression of genes for fat metabolism and cytokines. Subsequently, the cells were harvested for Western blot analysis and/or total RNA isolation.

In order to examine the effects of molecules released from RAW264.7 cells on the AML-12 cells, the conditioned medium was prepared from RAW264.7 cells. First, RAW264.7 cells were cultured with the control DMEM or LPS-contained DMEM in the absence of FBS for 24 h. Next, the culture supernatant was obtained by centrifugation at 3000× *g* for 3 min using conditioned medium. Finally, some AML-12 cells were incubated without a conditioned medium and served as the control. All groups were incubated with fresh media for 24 h and treated with sodium palmitate (PA, 250 mmol/L) or control in the presence of different concentrations of NaA (0.1, 2 mM) for the last 24 h and assessed for fat deposition.

In order to examine the direct effect of macrophage activation on hepatocyte fat deposition, AML-12 cells were incubated with RAW264.7 cells and assayed for hepatocyte fat deposition. Confluent RAW264.7 were seeded above transwell 12 well inserts with a pore size of 0.4 μm for incubation in DMEM. Once RAW264.7 monolayers were established, cells were washed once with phosphate-buffered saline (PBS). Cells in the CON group (control group) were incubated with fresh DMEM medium, while cells in NaA-L and NaA-H groups were incubated with DMEM or RPMI 1640 medium with 0.1 mM or 2 mM of NaA, respectively, for 24 h in the presence of LPS (20 ng/mL) for the last 6 h. Cells treated with LPS without NaA were used as the model group (MOD). Confluent AML-12 cells were seeded in DMEM with PA in lower transwell chambers so that RAW264.7 were suspended above AML-12 on the porous transwell membrane and stimulated with NaA and PA as mentioned above. Controls comprised RAW264.7 and AML-12 cultured alone in upper and lower transwell chambers. Lysates from lower transwell chambers were separately collected at 24 h [8,52]. The detailed process was included in Appendix A.

### 4.3. Cell Cytotoxicity Assays

An Enhanced Cell Counting Kit 8 Assay (Beyotime, Shanghai, China) was used to determine the cell cytotoxicity of NaA. RAW264.7 or Kupffer cells were seeded at a density of 5 × 10^3^ per well onto flat-bottom 96-well culture plates (Corning, Corning, NY, USA). Cells were treated with NaA (0, 0.01, 0.05, 0.1, 0.5, 1, 2, 5, or 10 mM). The absorbance values of viable cells were finally determined at 450 nm using a microplate spectrophotometer (BioTek, Winooski, VT, USA). The cell inhibitory rates were calculated using the following formula: Cell inhibition rate (%) = (1 − A_450_ (sample)/A_450_ (control)) × 100.

### 4.4. Western Blot

Total protein was isolated from cultured cells using a lysis buffer supplemented with protease and phosphatase inhibitors. The protein concentration was measured using a protein assay kit (Bio-Rad, Hercules, CA, USA). An equivalent of 30 μg protein extract was separated by sodium dodecyl sulfate-polyacrylamide gel electrophoresis (SDS-PAGE) and transferred to polyvinylidene difluoride membranes (PVDF). The membranes were incubated with the following primary antibodies: NF-κB p65, Pp65, p-c-Jun, c-Jun, β-Actin, pAMPK, AMPK, ACSS2, UCP2, PPARγ, pIκBα and IκBα (Abcam, Shanghai, China). After incubation with a goat anti-rabbit horseradish peroxidase-conjugated secondary antibody (Beyotime, Shanghai, China) at a dilution of 1:10,000 for 1 h, the proteins were finally visualized using a Luminata Forte Enhanced Chemiluminescence Kit (Millipore, Billerica, MA, USA) and detected by Imager 600 (Amersham, Switzerland). The band intensities were analyzed using ImageJ 1.54b (NIH).

### 4.5. Quantitative Real-time Polymerase Chain Reaction (qRT-PCR)

Total RNA was extracted using TRIzol Reagents (Ambion, Austin, TX, USA) and subjected to reverse transcription using a Prime Script RT-PCR kit (TaKaRa, Beijing, China). The qRT-PCR was carried out using SYBR Premix Ex Taq (TaKaRa, Beijing, China) on the Light-Cycler 480 (Roche Diagnostics) and analyzed by the LightCycler® 96 1.1 (Roche Diagnostics). The primers were synthesized by Sangon Biotech, China (Appendix A). The fluorescence data of the target genes were analyzed by the 2^−ΔΔCt^ method for relative quantification using Actin or GAPDH as an internal control.

### 4.6. Measurement of Intracellular Acetate Content

Sodium acetate extraction was performed as follows: 10^8^ cells were mixed with 2 mL extraction reagent (water: phosphoric acid = 1:3) and homogenized for 20 s at 6500× *g* using a Precellys 24 homogenizer (Bertin Technologies, Montigmyle Bretonnexux, France). The cell extract was prepared by adding 2 mL ether on ice for 10 min, followed by centrifugation at 4000× *g* for 20 min [53]. The remaining aqueous layer was further extracted with ether, after which the ether layers were pooled and diluted to 2 mL. Next, samples were subjected to gas chromatography-mass spectrometer (GC-MS) analysis using a 7890B gas chromatograph/5977 mass selective detector (Agilent Technologies, Santa Clara, CA, USA) with an HB-5 ms capillary column (30 m × 0.25 mm × 0.25 µm film thickness) (Agilent Technologies, Santa Clara, CA, USA ). Pure water was used as a blank sample to correct the background. A blank sample was processed similarly to that of fecal samples. The corrected peak area of acetate was calculated by the peak areas of samples minus that of the blank sample detected under the same conditions.

### 4.7. Flow Cytometry Analysis

RAW264.7 cells were detected by flow cytometry. Fc receptors of the above cells were first blocked with anti-mouse CD16/32 antibody (BioLegend, San Diego, CA, USA), after which the cells were respectively stained with fluorescent antibodies, including PE-conjugated anti-mouse CD86 antibody (BioLegend, San Diego, CA, USA), Brilliant Violet 421-conjugated anti-mouse CD206 (BioLegend, San Diego, CA, USA). The fluorescence antibodies were performed using intracellular staining as previously described [54]. Meanwhile, cells were respectively stained with isotype-matched control antibodies. Finally, prepared samples were measured and analyzed using a Cyto FLEX flow cytometer (Beckman Coulter, Brea, CA, USA).

### 4.8. Small Interfering (si) RNA Transfection

siRNA targeting *GPR43* or control siRNA were synthesized by Biolino Biotech (Tianjin, China) (Appendix A). Transfections were performed using the Lipofectamine^®^ 2000 RNAiMax reagent (Invitrogen, Karlsruhe, Germany) following the manufacturer’s instructions. As previously described, cells were treated with different concentrations of NaA and LPS for 24 h after 24 h post-transfection. qRT-PCR confirmed the downregulation of the *GPR43* targeted by siRNA.

### 4.9. Measurement of Intracellular AMP:ATP Ratio

Following exposure to sodium acetate, RAW264.7 cells were washed three times with ice-cold PBS, and intracellular nucleotides were extracted by adding 3 mL of ice-cold aqueous acetonitrile (50%, *v*/*v*) (VWR, Radnor, PA, USA) to cells. The resulting suspension was maintained on ice for 10 min, followed by centrifugation at 14,000× *g* for 1 min at 0 °C. Next, the supernatant was collected and dried using a refrigerated Savant SpeedVac vacuum concentrator (Thermo Fisher Scientific, Waltham, MA USA), after which the dried extract was resuspended in 240 μL of deionized water and filtered using a 0.22 μm syringe filter unit for high-performance liquid chromatography (HPLC) analysis [45]. The chromatographic separation and analysis were performed on an Agilent system (1200 series) equipped with a diode-array detector and a C18 reverse-phase column (Kromasil, 5 μm, 100 Å; 4.6 × 150 mm) at a flow rate of 1 mL/min and a linear gradient of acetonitrile (0~7%) in 10 mM triethylammonium acetate buffer (Glen Research, Sterling, VA, USA) over 20 min. AMP and ATP were identified based on their retention times.

### 4.10. Biochemical Assays

The levels of TG and TC were quantified using a triglyceride assay kit and a total cholesterol assay kit, respectively, according to the manufacturer’s protocols (Jiancheng, Nanjing, China). Hepatocyte lipid accumulation was analyzed using Oil Red O staining (Solarbio, Beijing, China), and the colorimetric assay was used to quantify triglyceride content.

### 4.11. Statistical Analyses

All experiments were conducted in parallel at least three times, and the data were presented as the mean ± the standard deviation (SD). The normality of the data was checked using the Kolmogorov–Smirnov test. One-way analysis of variance (ANOVA) and LSD post-tests were used to analyze the data after it passed the normality test by SPSS 20.0(Norman Nie, Chicago IL, USA). Differences were considered significant at a two-tailed *p*-value less than 0.05.

## 5. Conclusions

In this study, we found for the first time that NaA has a bidirectional modulatory effect on macrophage activity with dose-dependent effects, and high doses of NaA can inhibit macrophage inflammatory response and suppress hepatocyte fat accumulation based on macrophage-hepatocyte interactions. The macrophage AMPK activation state was found to be a key site of the bidirectional regulatory mechanism. The present study provides reliable evidence for the effective dose as well as a mechanism of NaA amelioration of NAFLD. It also revealed a bidirectional regulatory effect and dose effect of NaA on macrophage activity. Future validation should be performed in preclinical and clinical trials.

## Figures and Tables

**Figure 1 ijms-24-05536-f001:**
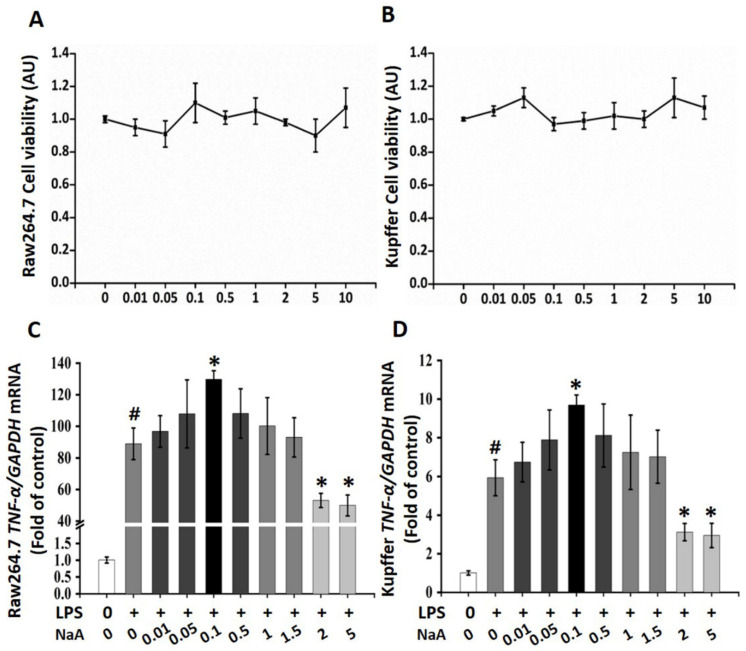
Effects of different doses of NaA on LPS-induced macrophage activation. NaA cytotoxic effects on RAW264.7 cells (**A**) or Kupffer cells (**B**). After serum starvation for 12 h, a CCK8 assay was performed for cells treated with 0~10 mM of NaA for 24 h. qRT-PCR was used to evaluate the mRNA expressions of *TNF-α* of RAW264.7 cells (**C**) or Kupffer cells (**D**). Cells were treated with or without 20 ng/mL LPS for 6 h following different doses of NaA for 24 h as detailed in the Materials and methods section. For all bar graphs, data are the mean ± SD, # *p* < 0.05, as compared with CON, and * *p* < 0.05, as compared with MOD. The significant difference was assessed using the one-way ANOVA followed by LSD post-tests. Control group (CON), group model cell treated with an LPS (MOD), arbitrary unit (AU).

**Figure 2 ijms-24-05536-f002:**
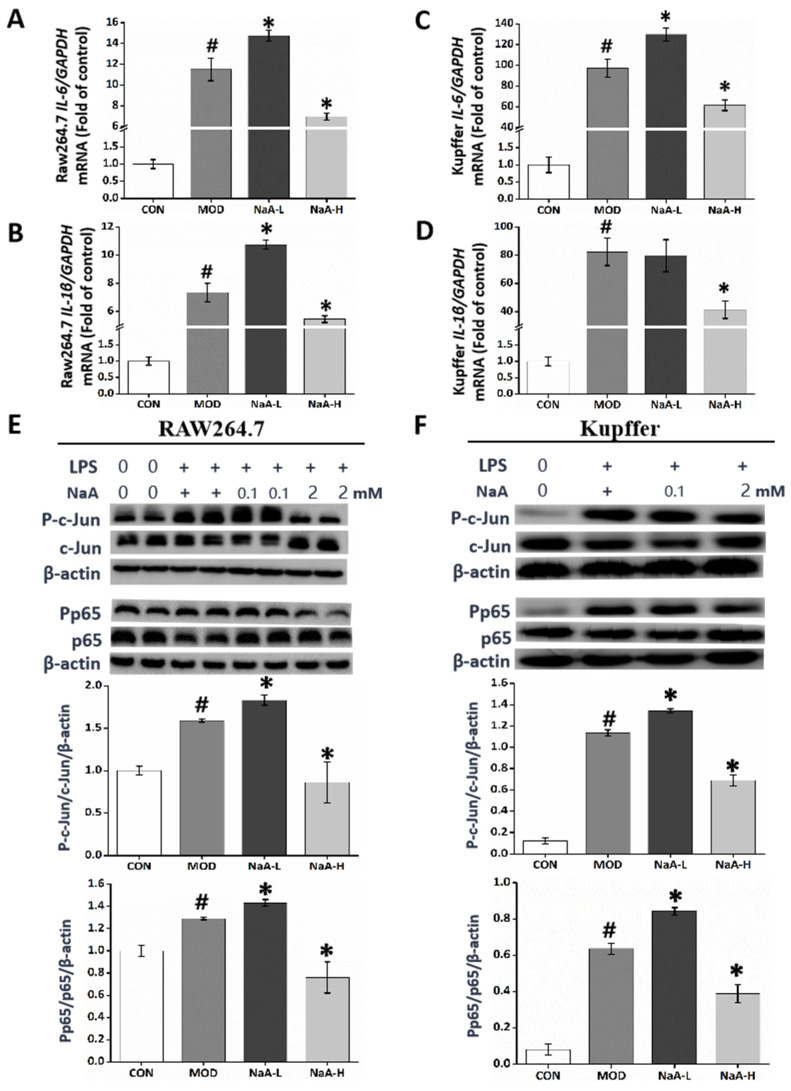
A high or low dose of NaA regulated LPS-induced macrophage activation with opposite effects. Cells were treated with or without 20 ng/mL LPS for 6 h and different doses of NaA for 24 h, as detailed in the Materials and methods section. Then qRT-PCR was used to evaluate the mRNA expressions of *IL-6, IL-1β* of RAW264.7 cells (**A**,**B**) and Kupffer cells (**C**,**D**). Cells were treated with or without 100 ng/mL LPS for 30 min and 0, 0.1, or 2 mM of NaA for 24 h. Then Western blot was used to evaluate the phosphorylation of c-Jun and p65 of RAW264.7 cells (**E**) and Kupffer cells (**F**). For all bar graphs, data are the mean ± SD, # *p* < 0.05, as compared with CON, and * *p* < 0.05, as compared with MOD. The significant difference was assessed using the one-way ANOVA followed by LSD post-tests. Control group (CON), group model cell treated with an LPS (MOD), 0.1 mM sodium acetate group (NaA-L), 2 mM sodium acetate (NaA-H), arbitrary unit (AU).

**Figure 3 ijms-24-05536-f003:**
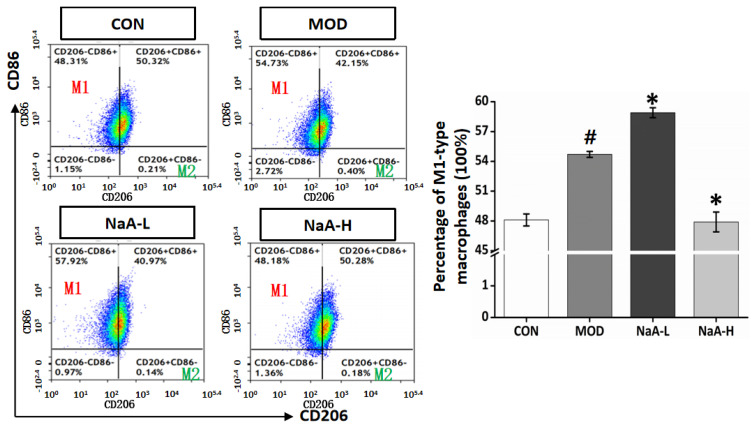
NaA alters the ratio of M1 macrophages in RAW264.7 stimulated by LPS. RAW264.7 cells were pretreated with NaA (0.1, 2 mM) for 18 h before treatment with LPS (100 ng/mL). Cells were harvested at 6 h of LPS incubation for further flow cytometry analysis. For all bar graphs, data are the mean ± SD, # *p* < 0.05, as compared with CON, and * *p* < 0.05, as compared with MOD. The significant difference was assessed using the one-way ANOVA followed by LSD post-tests. Control group (CON), group model cell treated with an LPS (MOD), 0.1 mM sodium acetate group (NaA-L), 2 mM sodium acetate (NaA-H), arbitrary unit (AU).

**Figure 4 ijms-24-05536-f004:**
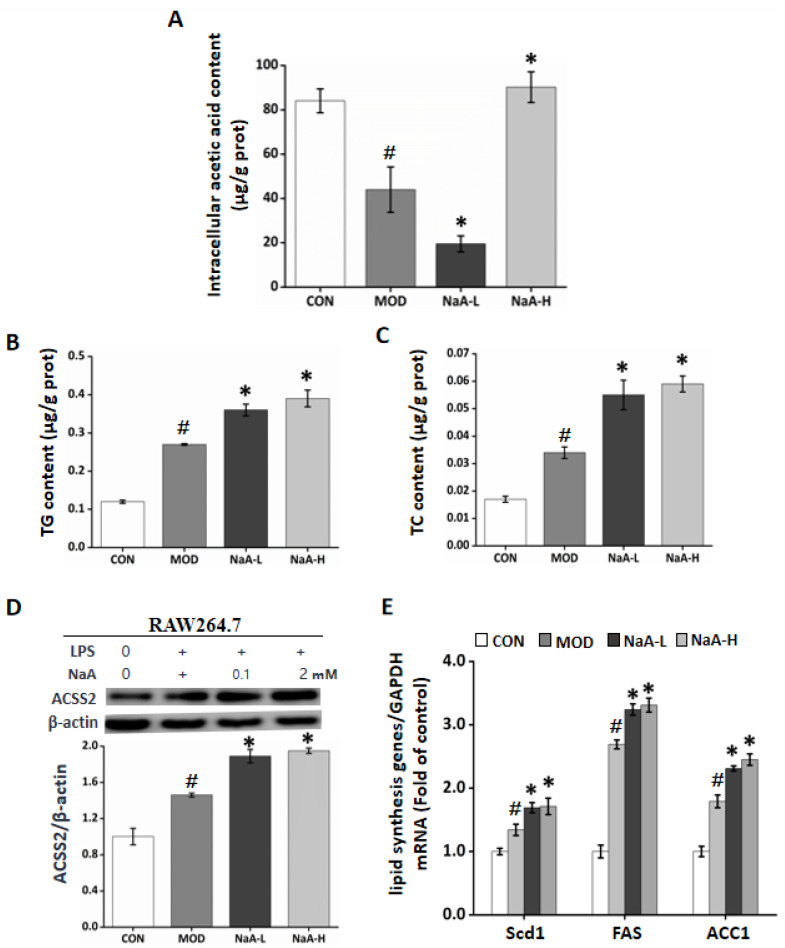
NaA alters intracellular free acetate content by promoting lipid synthesis in macrophages in a dose-related manner. (**A**) NaA regulated intracellular acetic acid content in a dose-dependent manner. (**B**) NaA increased intracellular TG content. (**C**) NaA increased intracellular TC content. (**D**) NaA enhanced the expression of ACSS2 protein, determined by Western Blot (**E**) qRT-PCR showed that NaA increased mRNA expression of lipid synthesis genes (*Scd1*, *FAS*, *ACC1*). For all bar graphs, data are the mean ± SD, # *p* < 0.05, as compared with CON, and * *p* < 0.05, as compared with MOD. The significant difference was assessed using the one-way ANOVA followed by LSD post-tests. Control group (CON), group model cell treated with an LPS (MOD), 0.1 mM sodium acetate group (NaA-L), 2 mM sodium acetate group (NaA-H), arbitrary unit (AU).

**Figure 5 ijms-24-05536-f005:**
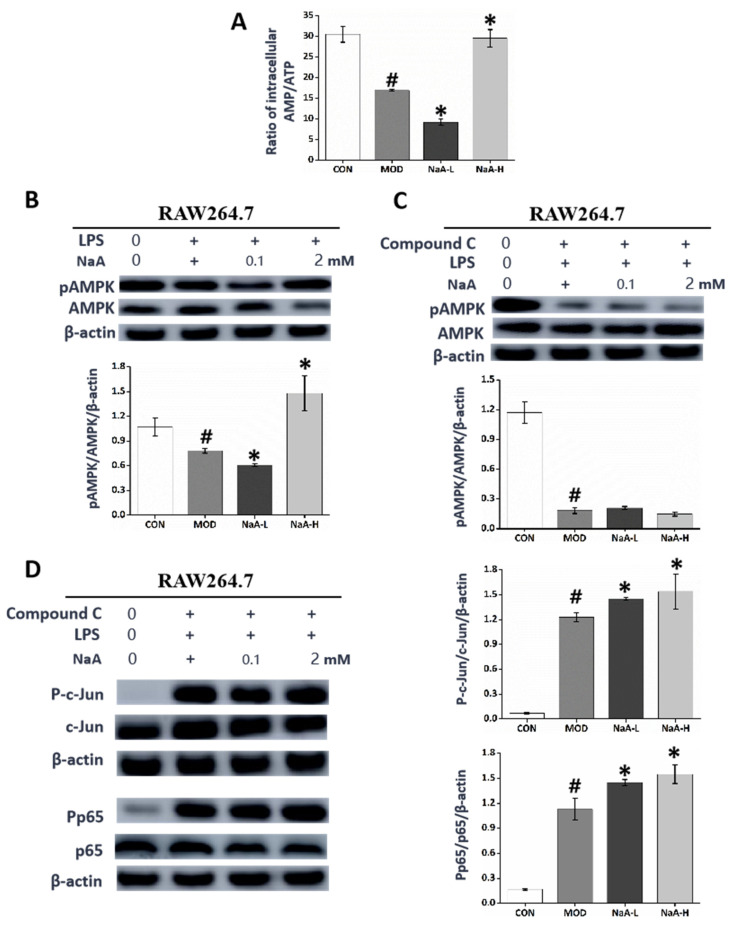
NaA-regulated macrophage inflammatory response is related to AMPK activity. (**A**) HPLC was used to evaluate the intracellular AMP/ATP ratio. (**B**) Western blot was used to evaluate the phosphorylation of AMPK. (**C**) Inhibitory effect of Compound C on AMPK. (**D**) Western blot was used to evaluate the phosphorylation of c-Jun and p65 treated by Compound C. For all bar graphs, data are the mean ± SD, # *p* < 0.05, as compared with CON, and * *p* < 0.05, as compared with MOD. The significant difference was assessed using the one-way ANOVA followed by LSD post-tests. Control group (CON), group model cell treated with an LPS (MOD), 0.1 mM sodium acetate group (NaA-L), 2 mM sodium acetate group (NaA-H), arbitrary unit (AU).

**Figure 6 ijms-24-05536-f006:**
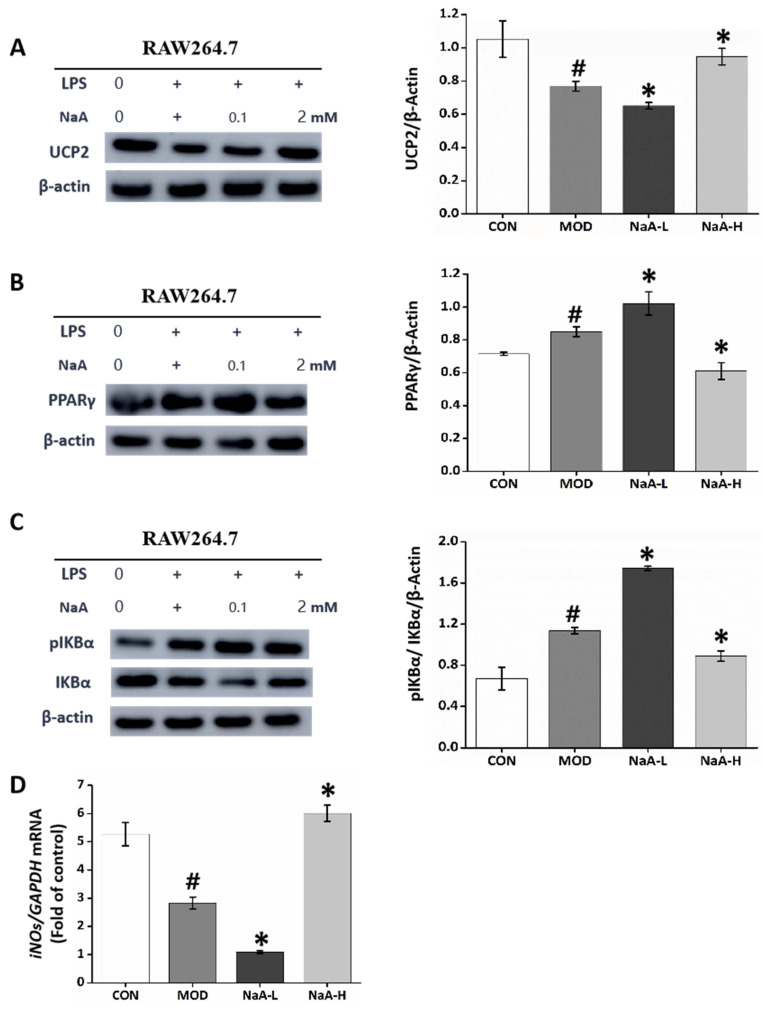
NaA affectsAMPK-related pathways to regulate inflammatory responses in macrophages. (**A**–**C**) Western blot was used to evaluate the expression of UCP2, PPARγ, and the phosphorylation of IKBα. (**D**) qRT-PCR was used to evaluate the mRNA expressions of *iNOS*. For all bar graphs, data are the mean ± SD, # *p* < 0.05, as compared with CON, and * *p* < 0.05, as compared with MOD. The significant difference was assessed using the one-way ANOVA followed by LSD post-tests. Control group (CON), group model cell treated with an LPS (MOD), 0.1 mM sodium acetate group (NaA-L), 2 mM sodium acetate group (NaA-H), arbitrary unit (AU).

**Figure 7 ijms-24-05536-f007:**
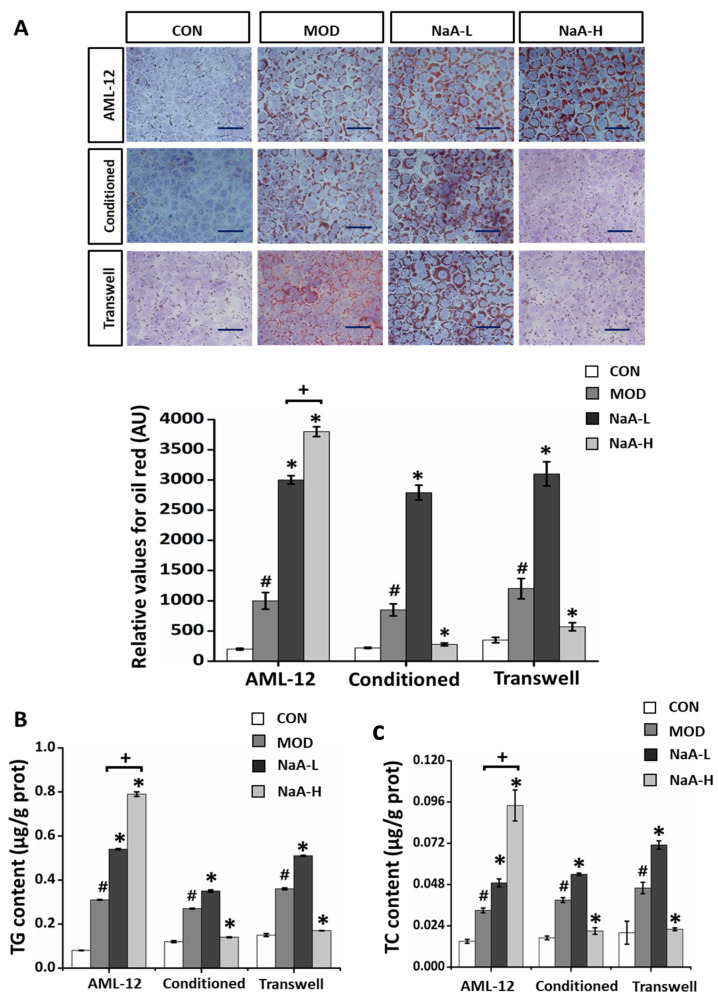
NaA affects the lipid accumulation of hepatocytes by regulating macrophage activity. (**A**) Oil Red O staining was used to detect the lipid deposition (magnification: 100×, scale bar: 500 μm). The bar graph displays percentages of lipid accumulation (*n* = 6–10). (**B**) The levels of hepatic triglycerides. (**C**) The levels of hepatic cholesterol. (**D**–**F**) qRT-PCR was used to evaluate the mRNA expressions of lipid synthesis genes (*Scd1, FAS, ACC1, SREBP, FASn*). For all bar graphs, data are the mean ± SD, # *p* < 0.05, as compared with CON, * *p* < 0.05, as compared with MOD, *+ p* < 0.05. The significant difference was assessed using the one-way ANOVA followed by LSD post-tests. Control group (CON), group model cell treated with an LPS (MOD), 0.1 mM sodium acetate group (NaA-L), 2 mM sodium acetate group (NaA-H), arbitrary unit (AU).

## Data Availability

Not applicable.

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
