# Peer review of "Bidirectional Regulation of Sodium Acetate on Macrophage Activity and Its Role in Lipid Metabolism of Hepatocytes"

_ijms, 2023, doi:10.3390/ijms24065536_

Round 1

Reviewer 1 Report

Remarks to the Author:

The author showed low or high dose of NaA has different effects on inflammation. However, it has been reported that NaA can decrease inflammation by reducing MAPK/NF-kB signaling. In-vitro results can not make a conclusion that NaA has a role on NAFLD. It seems that NaA influence many pathways on macrophage including MAPK, NF-kB, lipid synthesis, AMPK. However, what is the main target of NaA? The authors indeed found something new that low dose of NaA can increase inflammation.

Major points:

1.    Fig1. The dose of NaA should from 0.01 to 5 on C and D, like A and B shows, to see lower dose of NaA increases the expression of TNF-a.

2.    Fig2. It seems no different of p-c-Jun and pp65 level in fig 2F?

3.    Fig3. LPS reduced the level of M2 of MOD and NaA-L group compare with CON group. Why? How about only NaA treatment without LPS stimulated.

4.    Fig5. The change of lipid accumulation induced by NaA can not reflect inflammation activation.  Where is Fig4?

5.    Fig6. Only LPS or NaA treatment group should be a better control in B and C.

6.    Fig8. I have no doubt about that NaA affects lipid accumulation, but how do you say it is due to macrophage?

Author Response

Dear Reviewer,

Thank you for handling our manuscript. We carefully read your comments and made responses accordingly. The manuscript has been revised by a professional language agency. We highlighted all the revisions by ‘track changes’ in the manuscript. We are looking forward to your further comments.

  1. The author showed low or high dose of NaA has different effects on inflammation. However, it has been reported that NaA can decrease inflammation by reducing MAPK/NF-kB signaling.

Response: Thank you for your comments. A previous study indicated that a certain concentration (200 mmol/L) of NaA could alleviate high-carbohydrate-diet-induced intestinal inflammation mainly by suppressing MAPK activation and NF-κB phosphorylation in fish [1]. However other study reported that NaA can exacerbate the inflammatory response by synthesizing lipids [2, 3]. These contradictory results suggested the intracellular signaling effects of acetate are pleiotropic and multivariate, and the mechanism responsible for NaA on inflammation is not fully clarified. In the present study, we demonstrated that NaA could bi-directionally regulate macrophage activity. High concentration of NaA can inhibit the inflammatory response of cells by activating AMPK, and low doses of sodium acetate can exacerbate the inflammatory response of cells by inhibiting AMPK activity. The results of this study can explain to some extent the inconsistent effect of sodium acetate on the inflammatory response.

  1. In-vitro results cannot make a conclusion that NaA has a role on NAFLD.

Response: Thank you for your comments. Our study revealed that different concentration of NaA bi-directionally regulated macrophage activity, which in turn bidirectionally regulated lipid accumulation in hepatocytes through macrophage-hepatocyte interactions. Our results may mechanistically explain the inconsistent effects of NaA on NAFLD in different previous studies. In the future, animal experiments and even human experiments can be used to verify the regulatory effect of sodium acetate on fat accumulation in liver.

  1. It seems that NaA influence many pathways in macrophage including MAPK, NF-kB, lipid synthesis, AMPK. However, what is the main target of NaA?

Response: Thank you for your comments. We found that AMPK is a major target of NaA in the regulation of inflammatory response of macrophage. We observed NaA could affect the intracellular acetate concentration, which further bi-directionally alter the ATP/AMP. When AMPK expression was inhibited, the effect of sodium acetate on macrophage activity disappeared. Therefore, the results of this study indicate that AMPK is the main target.

  1. The authors indeed found something new that low dose of NaA can increase inflammation.

Response: Thank you for your comments. We found that low concentrations of NaA (0.1 mM) can be used to synthesize lipids and reduce intracellular free acetate content, which in turn reduces intracellular AMPK phosphorylation activity and promotes the inflammatory response of macrophages. The opposite result was obtained with high concentrations of NaA (2 mM). This is the main finding of this study.

Major points:

  1. Fig1. The dose of NaA should from 0.01 to 5 on C and D, like A and B shows, to see lower dose of NaA increases the expression of TNF-α.

Response: Thank you for your comments. In fact, we did study the effect of 0.01 mM and 0.05 mM sodium acetate on the inflammatory response of macrophages. The 0.01 ~ 0.05 mM of NaA could slightly increase the express of TNF-α, but there were no significant changes between these groups and MOD group. We revised Fig1, added the description in the legend and revised the manuscript (Page 7, Line 292-294).

  1. Fig2. It seems no different of p-c-Jun and pp65 level in fig 2F?

Response: Thank you for your comments. We analyzed optical density ratios from the 3 independent western blot images and found significant differences by statistical analysis. The results confirmed that NaA can change the phosphorylation status of c-Jun and P65 mainly by changing the magnitude of the p-c-Jun/c-Jun and Pp65/p65 ratios, not just the level of p-c-Jun and pp65. The representative bands we chose to put in the figure are not clear enough. We revised the Fig2 (E, F).

  1. Fig3. LPS reduced the level of M2 of MOD and NaA-L group compare with CON group. Why? How about only NaA treatment without LPS stimulated.

Response: Thank you for your comments. Based on the FCM, we found that the percentage of M2 macrophages was too low (< 1%). The results confirmed that macrophages induced by LPS mainly showed M1 polarization, which was consistent with previous research results [4,5]. Nevertheless, we found that LPS could slightly increase the proportion of M2 cells, while sodium acetate treatment reduced the proportion of M2 cells. Because of its small change and low proportion, we did not analyze the changes in M2. We did not do the NaA treatment alone on macrophages, but we add LPS in Macrophage to simulate the its activation by low level inflammation during the occurrence of NAFLD [6,7]. On this basis we further investigated the role of NaA. So we added low and high concentrations of NaA to the MOD group (LPS-treated), the focus of attention in this study was to explore the effect of NaA on the inflammatory response of macrophages in the activated state. We used different concentrations of NaA to treat macrophages alone and tested the toxicity of NaA to macrophages. We found that NaA did not affect the viability of macrophages within the experimental concentration range.

  1. Fig5. The change of lipid accumulation induced by NaA can not reflect inflammation activation. Where is Fig4?

Response: We apologized for the error in the layout and dropping Figure 4. We moved Fig4 ((A) GPR43 was not involved in macrophages inflammatory response regulated by NaA. (B) HDACs was not involved in macrophages inflammatory response regulated by NaA) into supplementary and merged Fig4 and Fig5 as the new Fig4. The underlying mechanism for the development and progression of NAFLD is complex and multifactorial. Different theories have been formulated, leading initially to the ‘two hits hypothesis’. According to this, hepatic accumulation of lipids secondary to sedentary lifestyle, high fat diet, obesity and insulin resistance, acts as the first hit, sensitizing the liver to further insults acting as a ‘second hit’. The ‘second hit’ activates inflammatory cascades and fibrogenesis [8]. Sources of lipid influx contributing to NAFLD development include lipid biosynthesis [9]. This suggests that lipid accumulation can cause cells to be more sensitive and susceptible to inflammatory responses. Previous studies also found that macrophage fat accumulation promoted its inflammatory response [10,11]. Our study also observed lipid accumulation was related to inflammatory response of macrophage especially under the treatment of the low concentration of NaA. We’ve added this section to the discussion. (Page 25, Line 557-559)

  1. Fig6. Only LPS or NaA treatment group should be a better control in B and C.

Response: Thank you for your comments. In the MOD group, the macrophage was treated by LPS alone to simulate the activation of macrophages by low level inflammation during the occurrence of NAFLD. On this basis we investigated the role of NaA to the MOD group. The results clearly revealed that low and high concentrations found a bidirectional regulation of AMPK phosphorylation activation, and confirmed that AMPK is an important target of NaA. We used different concentrations of NaA to treat macrophages alone and tested the toxicity of NaA to macrophages. We found that NaA did not affect the viability of macrophages within the experimental concentration range.

  1. Fig8. I have no doubt about that NaA affects lipid accumulation, but how do you say it is due to macrophage?

Response: Thank you for your comments. Macrophages play an important role in hepatic fat accumulation, and NAFLD/NASH is accompanied by macrophage infiltration and inflammatory activation observed in both humans and animals [12-14]. Previous studies indicated that hepatic infiltration of macrophages is the major cause of inflammatory responses in the pathology of NAFLD/NASH [15,16]. Activated macrophages produced pro-inflammatory cytokines such as TNF-α, IL-6, and IL-1β that severely interfere with hepatocyte fat metabolism and exacerbate fat accumulation [12,16], acting as the source of ‘second hit’ [8]. Our previous study found that NaA could alleviate MCD-induced NASH in mice, in which the response of macrophages to NaA is stronger than that of hepatocytes. The study confirmed that macrophages was the main target cells of NaA in NASH alleviation [6]. In this study, both low and high doses of NaA increased hepatocyte and macrophage fat accumulation. But only high-dose NaA, inhibited macrophage activity, consequently suppressed fat overaccumulation in hepatocytes through macrophage-hepatocyte interactions. The results clearly indicate that macrophages play an important role in inhibition of fat accumulation in hepatocytes by NaA. We clarify the role of macrophages in the discussion (Page 26, Line 621-624).

References:

  1. Li, M.; Hu, F.C.; Qiao, F.; Du, Z.Y.; Zhang, M.L. Sodium acetate alleviated high-carbohydrate induced intestinal inflammation by suppressing MAPK and NF-κB signaling pathways in Nile tilapia (Oreochromis niloticus). Fish Shellfish Immunol 2020, 98, 758-765, doi:10.1016/j.fsi.2019.11.024.
  2. Bulusu, V.; Tumanov, S.; Michalopoulou, E.; van den Broek, N.J.; MacKay, G.; Nixon, C.; Dhayade, S.; Schug, Z.T.; Vande Voorde, J.; Blyth, K., et al. Acetate recapturing by nuclear acetyl-CoA synthetase 2 prevents loss of histone acetylation during oxygen and serum limitation. Cell Rep 2017, 18, 647-658, doi:10.1016/j.celrep.2016.12.055.
  3. Mashimo, T.; Pichumani, K.; Vemireddy, V.; Hatanpaa, K.J.; Singh, D.K.; Sirasanagandla, S.; Nannepaga, S.; Piccirillo, S.G.; Kovacs, Z.; Foong, C., et al. Acetate is a bioenergetic substrate for human glioblastoma and brain metastases. Cell 2014, 159, 1603-1614, doi:10.1016/j.cell.2014.11.025.
  4. Shapouri-Moghaddam, A.; Mohammadian, S.; Vazini, H.; Taghadosi, M.; Esmaeili, S.A.; Mardani, F.; Seifi, B.; Mohammadi, A.; Afshari, J.T.; Sahebkar, A. Macrophage plasticity, polarization, and function in health and disease. J Cell Physiol 2018, 233, 6425-6440, doi:10.1002/jcp.26429.
  5. Sica, A.; Invernizzi, P.; Mantovani, A. Macrophage plasticity and polarization in liver homeostasis and pathology. Hepatology 2014, 59, 2034-2042, doi:10.1002/hep.26754.
  6. Deng, M.; Qu, F.; Chen, L.; Liu, C.; Zhang, M.; Ren, F.; Guo, H.; Zhang, H.; Ge, S.; Wu, C., et al. SCFAs alleviated steatosis and inflammation in mice with NASH induced by MCD. J Endocrinol 2020, 245, 425-437, doi:10.1530/joe-20-0018.
  7. Cobbina, E.; Akhlaghi, F. Non-alcoholic fatty liver disease (NAFLD) - pathogenesis, classification, and effect on drug metabolizing enzymes and transporters. Drug Metab Rev 2017, 49, 197-211, doi:10.1080/03602532.2017.1293683.
  8. Peverill, W.; Powell, L.W.; Skoien, R. Evolving concepts in the pathogenesis of NASH: beyond steatosis and inflammation. Int J Mol Sci 2014, 15, 8591-8638, doi:10.3390/ijms15058591.
  9. Kang, H.; You, H.J.; Lee, G.; Lee, S.H.; Yoo, T.; Choi, M.; Joo, S.K.; Park, J.H.; Chang, M.S.; Lee, D.H., et al. Interaction effect between NAFLD severity and high carbohydrate diet on gut microbiome alteration and hepatic de novo lipogenesis. Gut Microbes 2022, 14, 2078612, doi:10.1080/19490976.2022.2078612.
  10. Wu, H.M.; Ni, X.X.; Xu, Q.Y.; Wang, Q.; Li, X.Y.; Hua, J. Regulation of lipid-induced macrophage polarization through modulating peroxisome proliferator-activated receptor-gamma activity affects hepatic lipid metabolism via a Toll-like receptor 4/NF-κB signaling pathway. J Gastroenterol Hepatol 2020, 35, 1998-2008, doi:10.1111/jgh.15025.
  11. Govaere, O.; Petersen, S.K.; Martinez-Lopez, N.; Wouters, J.; Van Haele, M.; Mancina, R.M.; Jamialahmadi, O.; Bilkei-Gorzo, O.; Lassen, P.B.; Darlay, R., et al. Macrophage scavenger receptor 1 mediates lipid-induced inflammation in non-alcoholic fatty liver disease. J Hepatol 2022, 76, 1001-1012, doi:10.1016/j.jhep.2021.12.012.
  12. Cai, Y.; Li, H.; Liu, M.; Pei, Y.; Zheng, J.; Zhou, J.; Luo, X.; Huang, W.; Ma, L.; Yang, Q., et al. Disruption of adenosine 2A receptor exacerbates NAFLD through increasing inflammatory responses and SREBP1cactivity. Hepatology 2018, 68, 48-61, doi:10.1002/hep.29777.
  13. Huang, W.; Metlakunta, A.; Dedousis, N.; Zhang, P.; Sipula, I.; Dube, J.J.; Scott, D.K.; O'Doherty, R.M. Depletion of liver Kupffer cells prevents the development of diet-induced hepatic steatosis and insulin resistance. Diabetes 2010, 59, 347-357, doi:10.2337/db09-0016.
  14. Xu, H.; Li, H.; Woo, S.L.; Kim, S.M.; Shende, V.R.; Neuendorff, N.; Guo, X.; Guo, T.; Qi, T.; Pei, Y., et al. Myeloid cell-specific disruption of Period1 and Period2 exacerbates diet-induced inflammation and insulin resistance. J Biol Chem 2014, 289, 16374-16388, doi:10.1074/jbc.M113.539601.
  15. Wen, Y.; Lambrecht, J.; Ju, C.; Tacke, F. Hepatic macrophages in liver homeostasis and diseases-diversity, plasticity and therapeutic opportunities. Cell Mol Immunol 2021, 18, 45-56, doi:10.1038/s41423-020-00558-8.
  16. Luo, X.; Li, H.; Ma, L.; Zhou, J.; Guo, X.; Woo, S.L.; Pei, Y.; Knight, L.R.; Deveau, M.; Chen, Y., et al. Expression of STING Is Increased in Liver Tissues From Patients With NAFLD and Promotes Macrophage-Mediated Hepatic Inflammation and Fibrosis in Mice. Gastroenterology 2018, 155, 1971-1984.e1974, doi:10.1053/j.gastro.2018.09.010.

Reviewer 2 Report

This manuscript by Weiwei Li aimed to evaluate the mechanism of the dose effect of sodium acetate (NaA) on the regulation of macrophage activity, an implication to the progression of non-alcoholic fatty liver disease (NAFLD). This study examined the effects of high and low doses of NaA on macrophage activity in model cell lines, RAW264.7, or Kupffer cells treated with lipopolysaccharide (LPS). The high and low amounts of NaA increased or decreased intracellular acetate concentration in macrophages, respectively, consistent with changes in regulated macrophage activity. LPS significantly increased the M1 polarization ratio of RAW264.7 or Kupffer cells. GPR43 and HDACs were not involved in the regulation of macrophage activity by NaA. Both high and low doses of NaA increased cell lipid synthesis and accumulation in macrophages and hepatocytes, demonstrating that the inhibition of macrophage inflammatory response by NaA may depend on other pathways. NaA regulates the intracellular AMP/ATP ratio and AMPK activity in which the PPARγ/UCP2/AMPK/iNOS/IκBα/NF-κB signaling pathway plays a role. The results revealed that the macrophage AMPK activation by NaA is an important site of the bidirectional regulatory mechanism. Crosstalk between macrophage and hepatocyte inhibited macrophage inflammatory response and suppressed hepatocyte fat accumulation in response to high acetate doses.

A minor revision is required before being recommended for publication. See comments below:

Comment 1: The abstract needs vast improvement. Rewrite so that the study gap and objective are clear and results, discussion, and conclusion are coherent.

Comment 2: Where is figure 4? I could not locate it in the manuscript.

Comment 3: What is TC/TG content or TC/TG kit? Is it mean Plasma total cholesterol (TC) and triglycerides (TG)? If so, please insert it somewhere.

Comment 4: Section 3.4 in Lines 315-318 is very crucial. What did you say in lines 318-320? Did you say that the low expression of GPR43 in RAW264.7 is due to the silencing with the three siGPR43? If so, why did you conclude in line 321 by saying, “ This indicated that NaA-induced activation of RAW264.7 was independent of GPR43.”

Please explain what you meant by this sentence.

Author Response

Dear Reviewer,

Thank you for handling our manuscript. We carefully read your comments and made responses accordingly. The manuscript has been revised by a professional language agency. We highlighted all the revisions by ‘track changes’ in the manuscript. We are looking forward to your further comments.

Comment 1: The abstract needs vast improvement. Rewrite so that the study gap and objective are clear and results, discussion, and conclusion are coherent.

Response: Thank you for your comments. We have revised the abstract following your advice. (Page 1, Line 12-40).

Comment 2: Where is figure 4? I could not locate it in the manuscript.

Response: Thank you for your comments. We apologized for the error in the layout and dropping Figure 4. We moved Fig4 ((A) GPR43 was not involved in macrophages inflammatory response regulated by NaA. (B) HDACs was not involved in macrophages inflammatory response regulated by NaA.) to supplementary and merged Fig4 and Fig5 as the new Fig4.

Comment 3: What is TC/TG content or TC/TG kit? Is it mean Plasma total cholesterol (TC) and triglycerides (TG)? If so, please insert it somewhere.

Response: Thank you for your comments. We apologized for ambiguous description about the method. We used triglyceride assay kit and total cholesterol assay kit (Jiancheng, Nanjing, China) to determine the levels of TG and TC in the cultural cells. We have revised the manuscript (Page 7, Line 266-270).

Comment 4: Section 3.4 in Lines 315-318 is very crucial. What did you say in lines 318-320? Did you say that the low expression of GPR43 in RAW264.7 is due to the silencing with the three siGPR43? If so, why did you conclude in line 321 by saying, “ This indicated that NaA-induced activation of RAW264.7 was independent of GPR43.”

Please explain what you meant by this sentence.

Response: Thank you for your comments. We apologized for ambiguous description. GPR43 was considered as a receptor for acetate, so we used siRNA targeting GPR43 to verify its role in regulation of macrophage by NaA. We found that the background expression of GPR43 gene was low in RAW cells, and the gene silence did not affect the role of NaA. So we conclude that NaA-induced activation of RAW264.7 was independent of GPR43. We have revised the manuscript to make it clear. (Page 6, Line 346-348)

Reviewer 3 Report

The authors have conducted a critical study and presented their findings very well. However, there are some suggestions and questions to increase the scientific soundness of the manuscript.

Abstract:

1.     Why authors used “or” and have not used “and”? refer to line 20. If the findings were present in both macrophages and Raw 264.7 cells, then it is required to mention them explicitly.

2.     Authors must use the whole term or name before using an abbreviation. Refer to TG/TC line 24.

3.     Keywords can better be improved i.e., Sodium Acetate instead of acetate, dose-dependent effect instead of dose-depend effect.

Introduction

1.     According to a recent study published in “Lancet”, the worldwide incidence of NAFLD is more than 32%, not 25% (refer to line 37-38). The authors should update the figures. Authors should also check and verify other statistics. DOI:https://doi.org/10.1016/S2468-1253(22)00165-0

2.     Refer to lines 50-51; authors must provide the complete form of the abbreviation.

3.     Please provide the references to the claim “Accumulating evidence show that there is a close relationship between the gut and 62 NAFLD” (line 62-63)

Material and methods

1.     Correct “line” with “cell line” (line 110)

2.     Authors need to clarify whether they used “cell lines” or mouse-derived “primary cells”, in the case of Kupffer cells. (line 111-112).

3.     What are “CON” group and “other” groups? (line 136)

4.     How have the authors chosen the dose of palmitate 250 mmol/L? (line 149)

5.     For studying the effects of NaA on LPS-induced inflammation, RAW264.7 or 133 Kupffer cells authors used 6 different doses of NaA, while for examining the effects of molecules released from RAW264.7 cells on the AML-12 cells, only 2 concentrations of NaA were used. Explain this.

6.     At what concentration was LPS used? (line 156). It is better to mention rather than “As mentioned above” because there are 2 different doses of LPS provided in the upper paragraph.

7.     Why authors cultured hepatocytes and macrophages on trans-well inserts? (line 152-161) Normally, a co-culture is used.

8.     It is highly recommended that authors make a flow diagram or an illustration explaining their cell culture and drug/chemical dose treatment. Refer to lines 128 to 161.

9.     Write the exact doses of NaF “Cells were treated by NaA (0 ~ 10 mM).” (line 165)

10.  Why do authors perform the flowcytometry on RAW 264.7 cells? As they mentioned, “RAW264.7 cells were detected by flow cytometry”. Were authors not sure about the source of the cells/cell provider, i.e., Stem Cell Bank, Chinese Academy of Sciences (Shanghai, China)? Authors must mention the purpose.

Results

1.     Authors should decrease the number of figures by merging them.

Discussion

1.     Acetate or Sodium acetate? (line 485, 486) I think authors should use the name of their chemical of interest, “Sodium Acetate,” rather than acetate only.

Conclusion:

The authors should provide the implications of their study in the conclusion section.

Author Response

Dear Reviewer,

Thank you for handling our manuscript. We carefully read your comments and made responses accordingly. The manuscript has been revised by a professional language agency. We highlighted all the revisions by ‘track changes’ in the manuscript. We are looking forward to your further comment

The authors have conducted a critical study and presented their findings very well. However, there are some suggestions and questions to increase the scientific soundness of the manuscript.

Abstract:

  1. Why authors used “or” and have not used “and”? refer to line 20. If the findings were present in both macrophages and Raw 264.7 cells, then it is required to mention them explicitly.

Response: Thank you for your comments. We found that NaA had the same effect on both Kupffer and RAW 264.7 cells. We revised the word “or” to “and” following your advice. (Page 1, Line 20).

  1. Authors must use the whole term or name before using an abbreviation. Refer to TG/TC line 24.

Response: Thank you for your comments. We have revised the manuscript (Page 1, Line 43).

  1. Keywords can better be improved i.e., Sodium Acetate instead of acetate, dose-dependent effect instead of dose-depend effect.

Response: Thank you for your comments. We have revised the manuscript (Page 2, Line 52).

Introduction

  1. According to a recent study published in “Lancet”, the worldwide incidence of NAFLD is more than 32%, not 25% (refer to line 37-38). The authors should update the figures. Authors should also check and verify other statistics. DOI:https://doi.org/10.1016/S2468-1253(22)00165-0

Response: Thank you for your comments. We have revised the manuscript (Page 2, Line 59).(Page 2, Line 62)

  1. Refer to lines 50-51; authors must provide the complete form of the abbreviation.

Response:  Thank you for your comments. We have revised the manuscript (Page 2, Line 74).

  1. Please provide the references to the claim “Accumulating evidence show that there is a close relationship between the gut and 62 NAFLD” (line 62-63)

Response: Thank you for your comments. We added the references and revised the manuscript (Page 2, Line 85).

Material and methods

  1. Correct “line” with “cell line” (line 110)

Response:  Thank you for your comments. We have revised the manuscript (Page 3, Line 133).

  1. Authors need to clarify whether they used “cell lines” or mouse-derived “primary cells”, in the case of Kupffer cells. (line 111-112).

Response:  Thank you for your comments. We have revised the manuscript (Page 3, Line 134).

  1. What are “CON” group and “other” groups? (line 136)

Response:  Thank you for your comments. We apologized for ambiguous description. We have revised the manuscript (Page 4-5, Line 180-186).

  1. How have the authors chosen the dose of palmitate 250 mmol/L? (line 149)

Response:  Thank you for your comments. Previous studies used 250 mmol/L of palmitate to induce the fat accumulation in hepatocytes [1-3]. Our previous studies, as well as pre-experiments, have found that 250 mmol/L worked best (data not shown), so this concentration was employed in this study.

  1. For studying the effects of NaA on LPS-induced inflammation, RAW264.7 or 133 Kupffer cells authors used 6 different doses of NaA, while for examining the effects of molecules released from RAW264.7 cells on the AML-12 cells, only 2 concentrations of NaA were used. Explain this.

Response: Thank you for your comments. In fact, we did study the effect of 0.01 ~ 5 mM of NaA on the inflammatory response of macrophages. We clearly observed 0.01-0.1mM of NaA increased inflammation of RAW cells and Kupffer cells in dose- dependent manner, and 0.1 mM represented the strongest pro-inflammatory effects. However, 0.5 ~ 5 mM of NaA reduced the inflammatory response of macrophages. 2 mM and 5 mM of NaA ulteriorly suppressed the TNF-α expression compared with LPS-induced group (MOD), and no significant difference was observed between 2 mM and 5 mM NaA groups. Therefore, we chose 0.1 mM as the low-dose NaA group and 2 mM as the high-dose NaA group to analyze the mechanism of bi-directional regulating in activity of macrophage. We revised the Fig 1 in the manuscript.

  1. At what concentration was LPS used? (line 156). It is better to mention rather than “As mentioned above” because there are 2 different doses of LPS provided in the upper paragraph.

AU: Thank you for your comments. We apologized for ambiguous description about the method. We revised the manuscript (Page 4-5, Line 180-186).

  1. Why authors cultured hepatocytes and macrophages on trans-well inserts? (line 152-161) Normally, a co-culture is used.

AU: Thank you for your comments. We fully agree that co-culture and trans-well are common methods for studying cell interaction. The interactions between macrophages and hepatocytes are mainly dependent on cytokine signaling [2,3]. We explored whether different concentrations of NaA can directly regulate hepatocyte fat accumulation, or regulate hepatocyte fat accumulation through regulating macrophage activity bi-directionally. Both co-culture and trans-well are suitable methods. In addition to the interaction of cytokines, co-culture may also include the effect of direct contact between two kinds of cells [4]. In future studies, we can employ co-culture to further confirm that the effect of NaA depends on the regulation of macrophage activity.

  1. It is highly recommended that authors make a flow diagram or an illustration explaining their cell culture and drug/chemical dose treatment. Refer to lines 128 to 161.

AU: Thank you for your suggestion. We apologized for ambiguous description about the method. Following your advice, we revised the manuscript and prepared a flow diagram to illustrate the experiments, and put it in the supplementary (Fig S2)

  1. Write the exact doses of NaA “Cells were treated by NaA (0 ~ 10 mM).” (line 165)

AU: Thank you for your recommendations. We apologized for ambiguous description about the method. We revised the manuscript and made a flow diagram to illustrate the experiments. (Page 4, Line 162-163).

  1. Why do authors perform the flowcytometry on RAW 264.7 cells? As they mentioned, “RAW264.7 cells were detected by flow cytometry”. Were authors not sure about the source of the cells/cell provider, i.e., Stem Cell Bank, Chinese Academy of Sciences (Shanghai, China)? Authors must mention the purpose.

AU: Thank you for your comments. We chose RAW264.7 for flow cytometric assays and subsequent mechanistic probing. RAW264.7 is a well-established cell line for probing the inflammatory response of macrophages, which is more stable than Kupffer cells we used [5,6]. The flow cytometric assays were used to detect the state of the macrophages polarization. Macrophages commonly exist in two distinct subsets: 1) Classically activated or M1 macrophages, which are pro-inflammatory and polarized by LPS, and produce pro-inflammatory cytokines such as IL-1β, IL-6, IL-12, IL-23, and TNF-α; and 2) Alternatively activated or M2 macrophages, which are anti-inflammatory and immunoregulatory [7]. It is known that M1/M2 macrophage balance polarization governs the fate of inflammation [8]. So the results of flow cytometric assays could be used to analyze the effect on macrophage inflammatory response. We added the description of the purpose in the results (Page 13, Line 342-345).

Results

  1. Authors should decrease the number of figures by merging them.

AU: Thank you for your comments. Following to your suggestion, We moved Fig4 (((A) GPR43 was not involved in macrophages inflammatory response regulated by NaA. (B) HDACs was not involved in macrophages inflammatory response regulated by NaA.)) into supplementary and merged Fig4 and Fig5 as the new Fig4.

Discussion

  1. Acetate or Sodium acetate? (line 485, 486) I think authors should use the name of their chemical of interest, “Sodium Acetate,” rather than acetate only.

AU: Thank you for your comments. We revised and uniformed all descriptions (sodium acetate or NaA) in the manuscript.

Conclusion:

The authors should provide the implications of their study in the conclusion section.

AU: Thank you for your comments. The present study revealed a bidirectional regulatory effect and dose-dependent effects of NaA on macrophage activity, but the mechanism was only analyzed at the cellular level. Future validation will be performed in animals or human trials. We have revised conclusion section. (Page 27, Line 684-688).

References:

  1. Huo, Y.; Guo, X.; Li, H.; Xu, H.; Halim, V.; Zhang, W.; Wang, H.; Fan, Y.Y.; Ong, K.T.; Woo, S.L., et al. Targeted overexpression of inducible 6-phosphofructo-2-kinase in adipose tissue increases fat deposition but protects against diet-induced insulin resistance and inflammatory responses. J Biol Chem 2012, 287, 21492-21500, doi:10.1074/jbc.M112.370379.
  2. Cai, Y.; Li, H.; Liu, M.; Pei, Y.; Zheng, J.; Zhou, J.; Luo, X.; Huang, W.; Ma, L.; Yang, Q., et al. Disruption of adenosine 2A receptor exacerbates NAFLD through increasing inflammatory responses and SREBP1cactivity. Hepatology 2018, 68, 48-61, doi:10.1002/hep.29777.
  3. Luo, X.; Li, H.; Ma, L.; Zhou, J.; Guo, X.; Woo, S.L.; Pei, Y.; Knight, L.R.; Deveau, M.; Chen, Y., et al. Expression of STING Is Increased in Liver Tissues From Patients With NAFLD and Promotes Macrophage-Mediated Hepatic Inflammation and Fibrosis in Mice. Gastroenterology 2018, 155, 1971-1984.e1974, doi:10.1053/j.gastro.2018.09.010.
  4. Miki, Y.; Ono, K.; Hata, S.; Suzuki, T.; Kumamoto, H.; Sasano, H. The advantages of co-culture over mono cell culture in simulating in vivo environment. J Steroid Biochem Mol Biol 2012, 131, 68-75, doi:10.1016/j.jsbmb.2011.12.004.
  5. Khatua, S.; Simal-Gandara, J.; Acharya, K. Understanding immune-modulatory efficacy in vitro. Chem Biol Interact 2022, 352, 109776, doi:10.1016/j.cbi.2021.109776.
  6. Mitra, S.; Anand, U.; Sanyal, R.; Jha, N.K.; Behl, T.; Mundhra, A.; Ghosh, A.; Radha; Kumar, M.; Proćków, J., et al. Neoechinulins: Molecular, cellular, and functional attributes as promising therapeutics against cancer and other human diseases. Biomed Pharmacother 2022, 145, 112378, doi:10.1016/j.biopha.2021.112378.
  7. Gordon, S.; Martinez, F.O. Alternative activation of macrophages: mechanism and functions. Immunity 2010, 32, 593-604, doi:10.1016/j.immuni.2010.05.007.
  8. Shapouri-Moghaddam, A.; Mohammadian, S.; Vazini, H.; Taghadosi, M.; Esmaeili, S.A.; Mardani, F.; Seifi, B.; Mohammadi, A.; Afshari, J.T.; Sahebkar, A. Macrophage plasticity, polarization, and function in health and disease. J Cell Physiol 2018, 233, 6425-6440, doi:10.1002/jcp.26429.

Round 2

Reviewer 1 Report

  • You answered my questions.